# Retinoids as Chemo-Preventive and Molecular-Targeted Anti-Cancer Therapies

**DOI:** 10.3390/ijms22147731

**Published:** 2021-07-20

**Authors:** Victoria O. Hunsu, Caroline O. B. Facey, Jeremy Z. Fields, Bruce M. Boman

**Affiliations:** 1Center for Translational Cancer Research, Helen F. Graham Cancer Center & Research Institute, Newark, DE 19713, USA; vhunsu@udel.edu (V.O.H.); Caroline.Facey@christianacare.org (C.O.B.F.); 2Department of Biological Sciences, University of Delaware, Newark, DE 19713, USA; 3CA*TX Inc., Princeton, NJ 08540, USA; jzfields@comcast.net; 4Department of Pharmacology & Experimental Therapeutics, Thomas Jefferson University, Philadelphia, PA 19107, USA

**Keywords:** retinoic acid, cancer stem cells, adenomatous polyposis coli, aldehyde dehydrogenase, molecular targeted therapies

## Abstract

Retinoic acid (RA) agents possess anti-tumor activity through their ability to induce cellular differentiation. However, retinoids have not yet been translated into effective systemic treatments for most solid tumors. RA signaling is mediated by the following two nuclear retinoic receptor subtypes: the retinoic acid receptor (RAR) and the retinoic X receptor (RXR), and their isoforms. The identification of mutations in retinoid receptors and other RA signaling pathway genes in human cancers offers opportunities for target discovery, drug design, and personalized medicine for distinct molecular retinoid subtypes. For example, chromosomal translocation involving *RARA* occurs in acute promyelocytic leukemia (APL), and all-trans retinoic acid (ATRA) is a highly effective and even curative therapeutic for APL patients. Thus, retinoid-based target discovery presents an important line of attack toward designing new, more effective strategies for treating other cancer types. Here, we review retinoid signaling, provide an update on retinoid agents and the current clinical research on retinoids in cancer, and discuss how the retinoid pathway genotype affects the ability of retinoid agents to inhibit the growth of colorectal cancer (CRC) cells. We also deliberate on why retinoid agents have not shown clinical efficacy against solid tumors and discuss alternative strategies that could overcome the lack of efficacy.

## 1. Introduction

Our goal herein is to review the current research findings on retinoic acid (RA) agents, and to provide an update on clinical studies that use retinoids against cancers. We will firstly briefly discuss the function of the retinoid signaling pathway. Then, we will provide a discussion of retinoid drug structure, mechanism of action, and pharmacokinetics. Finally, we will discuss how the retinoid pathway genotype affects the ability of retinoid agents to inhibit the growth of colorectal cancer (CRC) cells. For example, given that different RA ligands have different affinities for the various RA receptor subtypes, it provides new strategies for designing molecular treatment approaches using select retinoid agents against cancers that carry mutations in specific RA receptor types. The overall focus of this paper is to understand how the identification of mutations in RA receptors and other RA signaling pathway genes in human cancers offers opportunities for target discovery, drug design, and personalized medicine, for distinct molecular retinoid subtypes. Moreover, we will make the case that recognizing why retinoid agents have not shown clinical efficacy against solid tumors provides insight into how to design alternative approaches that could overcome the lack of anti-cancer efficacy of retinoids.

## 2. Background

Retinoic acid (RA) is an active metabolite of vitamin A. It regulates a network of genes that are involved in embryonic development, cell growth, differentiation, tissue homeostasis, and apoptosis. Vitamin A is derived from plants and animals, and it has important functions in the human body. Mammals are able to synthesize vitamin A, which is obtained from beta-carotene-like colored fruit and vegetables, or animal sources, such as eggs and butter [1]. Retinoids are a family of signaling molecules, derived from retinol (vitamin A). In the body, retinoids are stored as retinyl esters in hepatic stellate cells [2]. They play a role in cellular differentiation, growth, and apoptosis [3]. Retinoids consist of a cyclic hydrophobic end group, a polyene or conjugated side chain, and a polar end group. Retinoids are small molecules, which allow them to enter cells through the plasma membrane. Retinoids are irreversibly oxidized to their naturally occurring RA derivatives (Figure 1 and Figure 2), which include all-trans retinoic acid (ATRA), 9-cis-retinoic acid (9-cis RA), and 13-cis-retinoic acid (13-cis RA). Synthetic retinoids, such as bexarotene [4] and fenretinide [5], are other available agents.

Retinoic acid has been shown to be a chemotherapeutic agent in the treatment of acute promyelocytic leukemia (APL). RA therapy has also been shown to improve the survival of neuroblastoma patients [6]. Using retinoids to treat other cancers has shown limited success. Nonetheless, the success of RA against APL has led to many investigations, to elucidate how RA signaling becomes dysregulated in cancer cells. For example, the mechanism underlying the ability of ATRA to induce differentiation of APL cells has been studied to understand how RA agents might be used for the differentiation of other cancer cell types. Indeed, preclinical studies show that 9-cis RA is useful in the prevention of prostate and breast cancer [7,8]. Moreover, 13-cis RA has shown clinical promise in patients with neuroblastoma and thyroid cancer [9]. The evidence for the activity of RA against other tumor types is much weaker. Information on retinoid signaling and different retinoid agents, as well as preclinical and clinical studies of RA and its dysregulation in various solid tumor types, bear further discussion (see below).

## 3. Retinoic Acid Signaling in Normal Cells

Retinoic acid is an active biological metabolite of vitamin A or all-trans retinol. RA and its related analogues (retinoids) regulate networks of genes that are involved in embryonic development, cell growth, differentiation, homeostasis, and apoptosis [10,11]. RA signaling (Figure 2) is mediated by the formation of a heterodimer, involving the following two retinoid receptor subtypes: retinoic acid receptor (RAR) and retinoic X receptor (RXR).

### 3.1. Retinoic Acid Receptors

The RARs and RXRs are members of a nuclear receptor family. Specifically, they are members of the ligand-dependent transcription factor family of steroid/thyroid/nuclear receptors [11,12,13,14]. Each receptor subtype has different isoforms. RARA and RARG have two isoforms, RARA1, RARG1, RARA2, RARG2, and RARB has five isoforms, RARB1-4 and B1′. All the subtypes of RXR have two isoforms, one and two [1,15]. The different receptor subtype genes are subjected to differential splicing and alternative promoters, which generate isoforms with distinct regulatory and functional properties [16].

The receptors are contained within the cytoplasm and sub-nuclear compartments, where they bind to RA and activate signaling through the formation of dimers, as reviewed in [17]. RAR interacts with RXR to form RAR/RXR heterodimers, which leads to the transcriptional regulation of primary genes after binding to retinoic acid response elements (RAREs) [14]. RAREs are comprised of tandem 5′-AGGTCA-3′ sites, known as the DR1–DR5 [18].

The receptors act as ligand-dependent transcription factors, which regulate the expression of primary target genes by specific binding to the RAREs on the promoter region of DNA [11,13]. The ligand ATRA binds to, and activates, RARs, with high affinity; the ligand 9-cis RA binds to, and activates, RXRs more than RARs, and the ligand 13-cis RA binds to, and activates, RARs. RARs are not only ligand-dependent transcriptional regulators, but can also exhibit extra-nuclear non-transcriptional effects and lead to activating kinase cascades within the nucleus [13]. RXRs form heterodimers with RARs, and can also form homo- and hetero-dimers with other nuclear receptors [1,19]. Depending on the partner, RXR can serve as a silent or active partner in the dimers. As a silent partner, RXR dimers will not regulate RA signaling, in which case RXR functions as a main regulator [1]. In the absence of a ligand, RAR/RXR heterodimers interact with a multiprotein complex that contains transcription co-repressors, which induce histone deacetylation, chromatin condensation, and transcription suppression [20]. In the presence of a ligand, the co-repressor is displaced from the receptors and the co-activators are recruited with histone acetyltransferases (HATs) activity, which leads to the activation of transcription [18,21].

### 3.2. Other Retinoid Signaling Components

The ability of RA to induce differentiation, cell cycle arrest, and apoptosis is also mediated through other cellular components, such as cellular retinoic acid-binding protein 1 (CRABP1) and fatty acid-binding protein 5 (FABP5) [22,23]. For example, RA has an enhanced apoptotic effect in cells that have a high CRABP1/FABP5 ratio. RA also binds to orphan nuclear receptors, nuclear receptor peroxisome proliferator-activated receptors (PPARB and PPARD), which increases the survival of cells that express high levels of FABP5 [24]. The RAR/RXR heterodimer acts as a transcriptional repressor in the absence of a ligand, and the RA receptors bind to DNA involving a co-repressor complex, nuclear co-repressor (NCoR) and silencing mediator for retinoid and thyroid receptor (SMRT) proteins. Both NcoR and SMRT form complexes with histone deacetylase (HDAC) [25,26]. The binding of RA alters the interactions of RAR/RXR heterodimers with co-regulatory proteins, and interacts with co-activator proteins with high affinity. Upon ligand binding, co-activator proteins bind to nuclear hormone receptors, to increase the transcriptional activity. The co-activator proteins include the p160/steroid receptor co-activator (SRC) protein family (SRC-1, SRC-2, SRC-3), with C-terminal domains that mediate the interaction with HATs, and coregulators such as (CBP/P300) [27,28,29]. Gillespie et al. [30] investigated cell lines that were mutated for the three different RARs, using chromatin immunoprecipitation (ChIP) analysis. This study revealed that the recruitment of co-activators to the RAR/RXR heterodimers occurs at specific RAREs sites. The binding of the RAR/RXR heterodimer to RAREs, and recruitment of coactivators, leads to the transcription of retinoid target genes, for further detail please see [30,31] (Figure 2).

## 4. Retinoid Agents

Below, we discuss the three known stereoisomers of RA, including all-trans retinoic acid (tretinoin), 13-cis retinoic acid (isotretinoin), and a less-stable isomer, 9-cis retinoic acid (alitretinoin). Other retinoid agents are inhibitors of the enzyme CYP26A1, which are collectively termed RAMBAs (retinoic acid metabolism blocking agents). Below, we also discuss the RAMBAs, liarozole and talarozole.

### 4.1. All-Trans Retinoic Acid (ATRA or Tretinoin)

#### 4.1.1. Mechanism of Action

All-trans retinoic acid is an active biological metabolite of vitamin A and has shown remarkable activity as a treatment for acute promyelocytic leukemia (APL) [32]. Classically, ATRA is a prime example of a cyto-differentiating agent that exerts its activity as an anti-tumor drug [33]. The pharmacological activity of ATRA is mediated through binding to RARA receptors [14,21]. It binds RARs with high affinity, but does not bind RXRs [34]. ATRA can be converted non-enzymatically to stereoisomers (9-cis RA and 13-cis RA) [35]. The mechanism of action (MOA) of ATRA is not fully clarified. However, the MOA following treatment of APL involves three processes. First, the restoration of the normal structure of the nuclear bodies, and the elimination of the PML–RARA protein via caspase-mediated cleavage and proteasome-dependent degradation. Second, the conversion of PML–RARA from a repressor to an activator. Third, the induction of gene expression by ATRA, which leads to differentiation [36]. Overall, ATRA induces differentiation of APL cells, because of its ability to promote PML–RARA degradation and dissociation of co-repressors [37].

#### 4.1.2. Pharmacological Uses

Due to the promising results of ATRA in the treatment of APL, several studies have been conducted to determine the effects of ATRA in the treatment of other diseases and cancers. In a recent study, Xiao-Jie Chen et al. investigated the effect of ATRA on cell proliferation, cell cycling, apoptosis, and PD-L1 (programmed death receptor ligand-1) expression in oral squamous cell carcinoma (OSCC) and oral dysplasia (OD) [38]. The study showed that ATRA induces anti-tumor effects by downregulating STAT3 expression, and subsequent signaling in OSCC and OD. The downregulation of STAT3 by ATRA caused inhibition of proliferation, induction of apoptosis, and downregulation of PD-L1 expression. The observed effects indicate that ATRA is a potential therapeutic agent against OSCC and OD [38].

In the case of breast neoplasms, ATRA induces re-differentiation of early transformed breast epithelial cells. A model of breast cancer progression was developed, by treating human normal-like breast epithelial cell lines with estradiol. While the cells at an advanced stage of progression did not show morphological changes following ATRA treatment, ATRA was able to re-differentiate the transformed cells at early stages of neoplastic progression, to a normal-appearing morphology. This finding suggests that ATRA may potentially be useful as a chemo-preventive agent, to prevent the progression of premalignant lesions of the breast [39]. As a differentiating agent, ATRA was found to inhibit cell proliferation, sphere formation, decrease stem cell (SC) population size, SC proliferation, and increase SC differentiation in ALDH+ colon cancer SCs [40]. Anti-tumor effects of ATRA were also observed in a study by Ming-Jer Young et al., whereby ATRA downregulated ALDH1 cancer SC (CSC) marker expression, and inhibited tumor formation in ovarian cancer cells [41]. The study confirms that ALDH1 is a CSC marker, and it regulates tumor formation in ovarian cancer cells through the downstream signaling of FoxM1/Notch1. Furthermore, ATRA downregulates ALDH1/FoxM1/Notch1 by targeting self-renewal pathways, and by suppressing sphere formation, cell migration, invasion of cells, and tumorigenesis [41].

### 4.2. 13-cis Retinoic Acid (Isotretinoin)

#### 4.2.1. Mechanism of Action

The 13-cis retinoic acid is a naturally occurring retinoid. Its use was first approved in 1982 as an oral treatment for severe acne, by the US Food and Drug Administration. It remains the only clinically effective therapy that is able to induce prolonged remission and significant improvement in acne patients [42,43]. The use of 13-cis RA has also been established as a treatment for children with high-risk neuroblastoma [44]. In neuroblastoma cell lines, 13-cis RA (Figure 1) has a longer half-life and higher, consistent peak plasma levels compared to other RA isomers. When administered orally, plasma concentrations peak between 2 and 4 h, and show an eliminated half-life of 10–20 h [45]. Also, 13-cis RA has different pharmacokinetic properties than ATRA and 9-cis-RA do [46], and is considered to be a storage form of biological active ATRA and 9-cis RA [47]. Further, 13-cis-RA does not demonstrate specific binding to RXRs and CRABPs, and it has a 100-fold lower binding affinity to RARs [48]. It can be converted intracellularly to metabolites, and act as a pro drug that is an agonist for RAR and RXR nuclear receptors [42,49]. Moreover, 13-cis RA is not active in transactivation assays, in the way that ATRA and 9-cis RA are, and this led to the general belief that it is not a key molecule in the regulation of gene transcription. The mechanism of action of 13-cis RA is unknown, but there are several possible mechanisms that explain the effects of 13-cis RA. First is the isomerization of 13-cis RA to ATRA [50]. The 13-cis RA may also regulate gene expression by binding to, as of yet, an unidentified nuclear receptor in an RAR–RXR independent pathway [51]. Second, 13-cis RA may be enzymatically isomerized, first to ATRA, and then to 9-cis RA [51]. So, 13-cis RA may be exerting its effects by regulating target gene transcription through ATRA or 9-cis RA [52,53]. Third, 13-cis RA may enhance cellular function by increasing target gene mRNA or its protein stability [54]. Fourth, 13-cis RA could be non-enzymatically produced by spontaneous isomerization from ATRA or 9-cis RA [52].

#### 4.2.2. Pharmacological Uses

A study by Matthay et al. investigated clinical outcomes in high-risk neuroblastoma pediatric patients, who were treated with chemotherapy, radiotherapy, transplantation, and 13-cis RA [55]. The response to treatment with 13-cis RA suggested that this retinoid was clinically beneficial for patients with improved outcomes, without progressive disease. This study suggested that while 13-cis RA was not effective in patients with high residual disease burden, it was effective in patients with the least residual disease, and was most effective in those that had a complete response [55].

In toxicity studies using animal models, the treatment of mice with 13-cis RA was associated with neurological effects, such as increased depression-related behavior, and impairment in spatial learning and memory [54]. These mood and behavior alterations were attributed to altered CNS function, from decreased hippocampal neurogenesis [56,57]. The effect on depression-related behaviors was not immediate, but required several weeks. Thus, any effect of 13-cis RA on gene expression must be a secondary effect accumulating over time, and not a direct, immediate influence on gene expression [58]. Fortunately, in humans, epidemiological studies have not reported a link between 13-cis RA use and clinical depression in patients.

An interesting study by Nelson et al. showed that 13-cis RA induced apoptosis and cell cycle arrest in human SEB-1 sebocytes by an RAR-independent mechanism [59]. The study evaluated the effects of ATRA, 9-cis RA, and 13-cis RA on the apoptosis, cell cycling, and proliferation in SEB-1 sebocytes and keratinocytes. Further, 13-cis RA produced greater effects at 48 and 72 h, in inducing apoptosis and inhibiting growth in sebocytes, compared to the other agents. The effects are attributed to the efficacy of 13-cis RA in reducing the production of sebum [59]. Thus, the induction of apoptosis appears to be the primary mechanism that explains the pharmacological mode of action and adverse effects of 13-cis RA in the treatment of neuroblastoma, APL, and severe acne [60].

### 4.3. 9-cis Retinoic Acid (Alitretinoin)

#### 4.3.1. Mechanism of Action

The 9-cis retinoic acid is an isomer of ATRA, but differs from ATRA due to its ability to activate both RAR and RXR. Unlike ATRA, which does not bind RXR, 9-cis RA is a high-affinity ligand for RXR. The activation of RXR by 9-cis RA allows for heterodimerization with RARs, peroxisome proliferator activator receptor (PPAR), thyroid hormone receptor (TR), vitamin D receptor (VDR), farnesoid receptor (FXR), pregnane X receptor (PXR), and constitutively activated receptor (CAR) [61]. Also, 9-cis RA can induce RXR homodimer formation, which elicits a response allowing for the activation of PPAR target genes. A study by Lin, Y. W et al. showed that 9-cis RA not only functions by regulating nuclear gene transcription, but it also regulates RXR in mitochondria, to activate mitochondrial gene transcription [62]. Moreover, 9-cis RA (0.1% gel) is approved by the FDA as a treatment for Kaposi’s sarcoma. Oral administration of 9-cis RA can be variably absorbed, but it is enhanced after food intake. The half-life is 2–10 h, and elimination occurs via excretion [63].

#### 4.3.2. Pharmacological Uses

A study by Yang et al. examined the effects of 9-cis RA on the growth and apoptosis of cutaneous T-cell lymphoma (CTCL) cells. The treatment of CTCL cells (HUT78, Myla) with 9-cis RA inhibited proliferation in both the cell lines and induced apoptosis. Furthermore, 9-cis RA treatment of these cell lines led to increased *RARA*, as well as increased *RARG* and *RXRA* retinoid receptor expression [64]. A preclinical study by Seong-Jin Yu indicated that 9-cis RA can induce neurological repair in the brains of adult male rats after stroke. To examine the neuro-reparative role of 9-cis RA in rats with stroke, 9-cis RA was administered intranasally, which induced repair of the brain and enhanced recovery of motor function. Thus, the authors concluded that 9-cis RA may offer a non-invasive treatment approach to patients with stroke [65].

### 4.4. Liarozole (R75251)

#### 4.4.1. Mechanism of Action

Liarozole is a derivative of imidazole and has two main mechanisms of action. First, it is an RA metabolism blocking agent (RAMBA) that inhibits the cytochrome P-450-mediated catabolism of RA [66]. The inhibition of the P-450 enzyme CYP26A1 leads to increased tissue and plasma levels of endogenous RA [67]. Second, it is an effective aromatase inhibitor [68,69].

#### 4.4.2. Pharmacological Uses

A study by Gilden Melissa et al. evaluated the influence of liarozole on the proliferation of leiomyoma fibrosis cells and the expression of extracellular matrix (ECM) components in immortalized leiomyoma cells [70]. The study showed that liarozole significantly decreased proliferation and ECM gene expression in leiomyoma cells [70]. Different doses of liarozole have been used in various studies, ranging from 50 mg to 300 mg. A multi-centered, double-blinded, placebo-controlled, dose-ranging study by Berth-Jones et al. determined the lowest effective oral dose of liarozole in the treatment of severe psoriasis vulgaris. A twice daily dose at 75 mg was the lowest effective dose [71]. The study also showed that oral liarozole is well-tolerated and provides an effective treatment for psoriasis.

Three phase II studies by P E Goss et al. were conducted to determine the efficacy and tolerability of liarozole in the treatment of estrogen receptor-negative (ER-), tamoxifen-refractory or chemotherapy-resistant postmenopausal metastatic breast cancer. Patient cohorts included ER-negative (group 1), ER-positive or unknown disease tamoxifen-refractory (group 2), ER-positive, and ER-negative chemotherapy-resistant (group 3) patients. The patients received oral doses (150–300 mg) of liarozole twice daily. The results showed that 25% of the patients in groups 1 and 2, and 11% of the patients in group 3 responded to the treatment. There was an 87% decrease in plasma estradiol levels after one month of treatment. The study results [72] also confirmed liarozole’s mechanism of action as a RAMBA. 

Another study by Verfaille et al. investigated the efficacy, safety, and tolerability of oral liarozole versus the second generation monoaromatic retinoid acitretin (etretin) in the treatment of patients with ichthyosis [73]. In this phase II/III study, the patients were randomized to receive either 75 mg of liarozole in the morning and evening (total 150 mg), or 10 mg of acitretin in the morning and 25 mg of acitretin in the evening (total 35 mg). The authors observed mild-to-moderate RA-related adverse effects in both the groups, but side effects occurred less frequently in the group that was treated with liarozole. Although this study shows that liarozole for ichthyosis is as effective as the retinoid acitretin, liarozole has better tolerability [73].

### 4.5. Talarozole (R115866)

#### 4.5.1. Mechanism of Action

Talarozole is another RAMBA that specifically inhibits CYP26 enzyme activity [69]. It increases endogenous RA in the plasma and skin, and exerts similar effects as those described for retinoids. Nanomolar concentrations of talarozole affect RA catabolism, and other CYPs are inhibited by micromolar concentrations. Talarozole is a more effective inhibitor of CYP26 (IC_50_ = 4 nM) compared to liarozole (IC_50_ = 3 uM) [74]. Moreover, talarozole has a greater specificity for CYP26 as compared to liarozole, with >300-fold higher IC_50_ values for CYP19, CYP17, CYP2C11, CYP3A, and CYP2A1 than for CYP26A1 [69].

#### 4.5.2. Pharmacological Uses

The pharmacological characterization of talarozole was reported in an animal study by Stoppie, et al. The rats were treated with a single dose of talarozole (2.5 mg/kg), which led to a significant increase in endogenous RA levels in the plasma, skin, fat, kidney, and spleen [74]. The RA levels returned to baseline after 18 h following administration. Treatment with talarozole showed typical retinoid side effects, which include a decrease in vaginal keratinization, induced pinnal hyperplasia, conversion of caudal para to orthokeratosis, and up-regulation of hepatic CYP26 expression in rats [74]. The effects of talarozole were studied in combination with ATRA on cultured human epidermal keratinocytes. Giltaire, S et al. showed that talarozole potentiates the effects of ATRA [75]. An effect from talarozole was not observed in the absence of ATRA. However, when talarozole was combined with low concentrations of ATRA, the effect of ATRA was potentiated. The effect was evident by the increased expression of HB-EGF and involucrin, and a decrease in keratin 10 expression [75].

In an open-label, single-arm trial, patients with moderate to severe plaque-type psoriasis were treated with talarozole (1 mg/day) for eight weeks, to assess efficacy, safety, and tolerability. The treatment resulted in a significant reduction in psoriasis, as measured by the severity index (PASI), with additional improvement seen even after the follow-up period. Orally administered talarozole appears to be well-tolerated, but further evaluation of efficacy and safety is ongoing [76]. For example, Verfaille et al. also confirmed the efficacy and tolerability of talarozole (1 mg/day for 12 weeks) in male patients with moderate-to-severe facial acne vulgaris [77]. While numerous studies have evaluated the anti-tumor effect of liarozole and another RAMBA, ketoconazole, the potentiality of talarozole as an anti-tumor agent has not yet been explored, indicating the need for further investigation. 

## 5. Effect of Retinoids in the Chemoprevention and Treatment of Various Cancers

There have been a number of clinical trials conducted on solid tumors, using retinoids, which we have listed in Table 1. Below, we discuss select preclinical studies and clinical trials that have been conducted in different tumor types.

### 5.1. Retinoids and Acute Promyelocytic Leukemia (APL)

Pharmacological doses of retinoids are useful in the treatment of some cancer types, and able to restore a normal response in the differentiation of tumor cells. A classic example is the use of ATRA in the treatment of APL, which is the most efficacious use of RA in cancer therapy since the mid-1980s. In 1995, the United States Food and Drug Administration approved the use of oral ATRA in patients with APL, following a report on the potentiality of ATRA to induce in vitro differentiation of APL cells and provide a therapeutic tool in the treatment of acute myeloid leukemia (AML) [78]. To date, the oral administration of ATRA is still considered a standard therapy in the treatment of APL. Moreover, APL is the only type of cancer that can reach 95% remission with ATRA and chemotherapy [79]. Studies have shown that most APL cases have a chromosome translocation involving chimeric fusion of promyelocytic leukemia genes and the *RARA* gene. The PML–RARA fusion protein acts as a transcriptional repressor that blocks physiological RA-mediated differentiation of hematologic precursors, which leads to the accumulation of granulocyte precursor promyelocytes [80]. In the treatment of APL, ATRA binds the RARA aspect of the fusion protein that leads to the dissociation of the co-repressors, which induces differentiation and complete remission in 95% of patients [81,82]. 

The combination of arsenic trioxide with ATRA has further improved the efficacy of treatment for APL. Arsenic trioxide binds to the PML aspect of the fusion protein and can even lead to a relatively high cure rate, as a single agent in APL patients [79,83]. Studies also show that the combined use of ATRA and arsenic trioxide act synergistically to enhance apoptosis and/or differentiation, depending on the dose [84]. The successful use of ATRA in the treatment of APL has led to efforts to test natural retinoids in the treatment of other cancers. Unfortunately, the therapeutic and chemo-preventive benefits of retinoids in solid tumors remain uncertain.

### 5.2. Retinoids and Breast Cancer

The first preclinical analyses of RA on breast cancer were done using cell lines, which dates back to the late 1970s. In these early preclinical studies, three of the breast cancer cell lines, Hs578T, 734B, SK-BR-3, showed growth inhibition, as follows: 27%, 50% and 83%, respectively. However, the MDA-MB-157 and Hs578Bst cell lines were not affected [85]. Fenretinide, a synthetic analog of retinoids, has also been studied in breast cancer prevention trials. An eight-year clinical study of fenretinide and breast cancer patients revealed no significant difference in the reduction in contralateral and ipsilateral breast cancer [86]. Some findings suggest that the RA receptor RARA and the estrogen receptor ER∝ target common genes throughout the genome, to antagonistically regulate the genes that are associated with breast cancer [87]. Consequently, a clinical trial was conducted to investigate the effect of adding hormonal therapy with retinoids in post-menopausal patients with advanced breast cancer. The efficacy of 13-cis-RA vs. interferon alpha2 (IFN∝2) with tamoxifen, showed no significant difference in the overall survival after eight years [88]. A phase II trial, performed by Bryan et al., investigated the drug combination of ATRA and paclitaxel in patients with recurrent or metastatic breast cancer, which revealed an overall clinical benefit of 76.4%. While the results appeared promising, the time to progression and survival rates were similar to those reported for paclitaxel alone. However, an ability of ATRA to induce differentiation and cell cycle arrest was evident, as increased stable disease was observed [89].

### 5.3. Retinoids and Prostate Cancer

For prostate cancer (Pca) patients, hormonal agents are typically used to treat recurrent disease following prostatectomy or radiotherapy, and for metastatic disease, but tumors often become hormone refractory [90]. The use of fenretinide as a single agent therapy in hormone refractory Pca patients with rising prostate-specific antigen (PSA), did not show a reduction in PSA, suggesting a need for drug combination-based studies [91]. For example, a randomized phase II study by Ferrari et al., involving patients with androgen-dependent prostate cancer, evaluated 13-cis RA in addition to hormonal therapy. The results showed that 13-cis RA did not lead to a decline in PSA, nor did it reduce hormonal therapy toxicity. Additionally, the antitumor activity of the drug combination was not superior to the standard hormonal treatment [92]. Although one study found that a loss of *RXRA* contributes to the tumorigenesis of prostate cancer [93], to date there are no clinical trials that provide convincing evidence for the effectiveness of retinoids in prostate cancer. 

### 5.4. Retinoids and Lung Cancer

A number of studies have been conducted to investigate the role of single-agent retinoids in patients at high risk for developing lung cancer, and those with a history of lung cancer. Early preclinical analyses on the effects of ATRA on human lung cancer cell lines showed that 17 of 22 human small-cell lung cancer (SCLC) cell lines, and 9 of 15 non-small-cell lung cancer (NSCLC) cell lines, were resistant to retinoids. Two NSCLC cell lines even showed growth stimulation following treatment with ATRA [94]. A randomized phase II study by Arrieta Oscar et al. evaluated the combination ATRA, with paclitaxel and cisplatin, in advanced NSCLC patients. The median progression-free survival (PFS) rate and response rate (RR) favored the incorporation of ATRA in patients, as indicated by the clinical outcome measures (RR—55.8% compared to 25.4%; PFS—8.9 months compared to 6 months) and acceptable toxicity [95]. A phase III study assessed the addition of bexarotene, a synthetic form of vitamin A, in first-line treatment with carboplatin and paclitaxel for NSCLC patients; however, there was no improvement in patient survival. Although, the occurrence of high-grade hypertriglyceridemia in bexarotene-treated patients correlated strongly with increased survival suggesting it may still hold some benefit in first-line treatment for NSCLC patients [96]. A meta-analysis looking at the use of naturally occurring retinoids in the treatment of lung cancer showed a lack of benefit; however, bexarotene may still hold promise for select patients [97]. Overall, these studies have not shown a clear benefit for use of retinoids in lung cancer.

### 5.5. Retinoids and Colorectal Cancer (CRC)

As discussed above, there have been a number of clinical trials conducted on solid tumors using retinoids, but we did not find any clinical trials listed on www.clinicaltrials.gov involving the treatment of CRCs with retinoid agents. However, a number of preclinical studies have been conducted on retinoids for CRC (discussed below), and the reader is also referred to a recent review on the subject [17].

### 5.6. Chemoprevention of Cancer Using Retinoids

Various studies have also been conducted to evaluate the chemo-preventive potential of retinoids as single agents, and in combination drug regimens with other chemotherapeutic agents. These studies were conducted on premalignant conditions, such as oral leukoplakia, erythroplakia, xeroderma pigmentosum, bronchial metaplasia, laryngeal papillomatosis, cervical dysplasia, actinic keratosis, and dysplastic nevi, as well as chemoprevention for second primary malignancies, such as esophageal, basal cell, breast, hepatocellular, squamous cell skin, cervical, and bladder carcinomas. The reader is referred to several excellent reviews on the subject [98,99,100,101,102,103].

## 6. Retinoid Drugs Are Not Efficacious Against Solid Tumors—Why Not?

Retinoic acid (RA) agents possess anti-tumor activity through their ability to induce cellular differentiation. Indeed, as discussed above, ATRA is a first-line treatment that cures many acute promyelocytic leukemia (APL) patients, mainly due to ATRA’s ability to induce the differentiation of APL cells into neutrophils [104,105,106]. Even though retinoids show promising pre-clinical activity in human solid tumors, they have not yet been translated into effective systemic treatments for most solid tumors. We believe that recognizing why retinoid agents have not shown clinical efficacy against solid tumors could provide insight into how to design alternative approaches that could overcome the lack of anti-cancer efficacy of retinoids.

To explore this line of thinking, here, we provide a deliberation on studies that have investigated the anti-cancer activity of retinoids in models of CRC [107,108,109], particularly intestinal tumor development in the *Apc^Min/+^* mouse model for familial adenomatous polyposis (FAP) patients who carry germline *APC* mutations [110].

One of the first things we considered, which might explain the lack of retinoid efficacy in colon tumors, was the presence of RA receptor mutations. Our search of RAR and RXR mutations rates in CRC shows that, while mutations occur in all of the different RA receptors, the mutation rate is quite low (<3%). The reader is directed to reference [17] to get more information about the RA receptor mutations in CRC. There is other evidence that the expression of some RA receptor genes (e.g., *RARB*) can be inactivated by DNA hypermethylation [11]. Indeed, decreased RA receptor signaling has been shown to occur in CRC [11,102,111]. Our bioinformatics investigation of the TCGA database [112] provides additional information—it showed that *RARA* and *RXRA* receptor levels are prognostic (*p* < 0.05) of CRC patient survival (Figure 3) [17]. Since RA receptor levels predict patient survival, this suggests that RA receptors might play a role in tumor growth.

However, other factors are likely to play a role in drug resistance to retinoids. For example, Volate et al. [107] showed that RA receptors, including *Rara, Rarb, Rxrb, Rxrg*, were expressed in *Apc^Min/+^* adenomas. Given the lack of receptor mutations in *Apc^Min/+^* mice, one would expect that ATRA should prevent adenoma formation in *Min* mice. However, Mollersen et al. [113] tested ATRA in *Apc^Min/+^* mice and found that ATRA treatment failed to prevent tumor formation. Other studies were then conducted to discover which mechanisms can explain this unexpected ATRA resistance in *Apc^Min/+^* mice.

A key study by Shelton et al. [114] provided an important clue, by analyzing the expression of *CYP26A1*, ATRA’s major catabolic enzyme. They found that *CYP26A1* expression was increased in tumors from *Apc^Min/+^* mice, and in tumors from FAP patients. They also determined that *CYP26A1* is a TCF4 target gene, which explains why CYP26A1 expression is increased due to upregulated WNT signaling in *Apc*-mutant tumor tissues. Thus, an increase in *CYP26A1*, due to increased ATRA degradation, provides a mechanism that explains the resistance of *Apc^Min/+^* mice to ATRA treatment, and the failure of ATRA to prevent tumor development.

A recent study by Penny et al. [110] provided more solutions to the problem. This study involved treating *Apc^Min/+^* mice with the CYP26A1 inhibitor liarozole. The administration of liarozole to *Apc^Min/+^* mice increased endogenous RA signaling by blocking ATRA metabolism, and dramatically reducing intestinal adenoma numbers in these *Apc*-mutant mice. We also found that the treatment of human CRC cells with liarozole decreases proliferation, sphere formation, and the size of the ALDH + SC population [115]. This suggests that decreasing the intracellular metabolism of ATRA, by inhibiting CYP26A1 activity using liarozole, might be a way to increase ATRA levels and augment RA signaling in tumor cells, including the reduction in tumor SC numbers in *APC*-mutant tissues.

These exciting results from pre-clinical studies raise the following important question: “what is the frequency of *APC* mutation in human cancer?” As it turns out, *APC* is one of the most frequently inactivated tumor suppressor genes across all solid tumor types (Figure 4) [116,117]. While inactivation of *APC* in FAP and sporadic CRCs is due to *APC* mutation, an often-unnoticed fact is that, in other cancers, *APC* inactivation often results from the hypermethylation of APC’s promoter, and its epigenetic silencing. Thus, if CYP26A1 is a TCF4 target gene that is upregulated by *APC* inactivation, it would increase ATRA degradation, lower intracellular ATRA levels, and diminish RA signaling. This provides a plausible mechanism that explains why retinoid drugs are not efficacious against solid tumors.

## 7. Pilot Study on the Effects of Retinoid Agents on *APC*-Mutant CRC Cells That Also Carry RA Receptor Mutations

We then performed a pilot study to analyze *APC*-mutant CRC cell lines that have mutations in the RA receptor genes, to determine if cells that carry RA receptor mutations show resistance to retinoid agents [118]. The results (Figure 5) show that HT29 cells, which only have wild-type RA receptor, are sensitive to ATRA and 13-cis RA compared to SW480 cells that have mutant RARA & RXRG, and HCT116 cells that have mutant RARA. LoVo cells, which carry a mutation in RXRA, have also been shown to be resistant to retinoid treatment [108]. Interestingly, all three cell lines we analyzed showed the same dose response to the CYP26A1 inhibitor liarozole, indicating that the inhibition of ATRA metabolism that increases intracellular RA levels has the same effect on CRC cells regardless of whether they carry RA receptor mutations. Another important aspect of this study was that all the CRC cells analyzed showed some response to the growth-inhibitory effects of retinoids, even though the cells carry *APC* mutations that would lead to WNT activation.

Indeed, the CRC cells with RA receptor mutations still showed growth inhibition at higher doses of RA ligands, which indicates that these cells are still responsive to RA signaling, but only if higher intracellular RA levels are achieved. Moreover, it shows that RA receptor mutant cells show the same dose response to the CYP26A1 inhibitor liarozole as non-mutant cells do; this indicates that the inhibition of CYP26A1 by liarozole leads to a high enough intracellular RA level to induce growth inhibition. Thus, we predict that liarozole and other RAMBAs will provide effective therapeutic approaches to target cancer cells via RA signaling, and treatment with liarozole could overcome the lack of efficacy that is traditionally seen in treating solid tumors with conventional retinoid agents.

## 8. The CYP26A1 Inhibitor Liarozole Decreases CRC Cell Proliferation, Sphere Formation, and Number of ALDH+ Cancer Stem Cells (CSCs)

Because the CYP26A1 inhibitor liarozole is able to decrease the proliferation of CRC cells (discussed above), we also treated CRC cells with liarozole, to determine the effects of CYP26A1 inhibition on sphere formation and SC numbers. We found that liarozole decreased the number of colonospheres that formed in a dose-dependent manner. Also, increasing doses of liarozole decreased the number of ALDH + CSCs [115]. Hence, our results indicate that liarozole-induced inhibition of CYP26A1 leads to a decrease in cell proliferation, SC self-renewal, and CSC numbers, by raising intracellular ATRA levels in CRC cells. Moreover, our recent studies show that a link between WNT signaling and retinoic acid signaling regulates colonic SCs, and human CRC evolves due to an imbalance between WNT and RA signaling [119]. These findings could have broad clinical significance, as most solid tumors are known to have the following: (i) SC overpopulation that drives tumor development and growth, and (ii) inactivated *APC* that, by upregulating CYP26A1 expression and increasing ATRA degradation [108], could induce resistance to retinoids and reduce the effectiveness of retinoid therapies.

Other mechanisms that become altered due to *APC* mutations may also contribute to the lack of efficacy of retinoid drugs against solid tumors. For instance, cancer growth is usually sustained by the overexpression of proteins from the inhibitor of apoptosis protein (IAP) family, such as survivin (BIRC5), which promotes cancer cell proliferation [120,121]. Indeed, survivin is highly overexpressed in many solid tumors, including lung, pancreatic, breast, and colorectal cancers [122,123]. Survivin plays a role in promoting cell division by activating aurora-B kinase, and inhibiting apoptosis by blocking caspase activation. Notably, we found that survivin is a TCF4 target gene. We observed that *APC* mutations lead to the up-regulation of survivin expression in neoplastic intestinal tissues in mouse and man [124,125]. We also reported, and others confirmed, that the expression of survivin is downregulated by beta-catenin/TCF-4 signaling, which is normally controlled by wild-type APC [126,127,128]. Moreover, we have shown that the chemo-preventive agent sulindac attenuates the expression of survivin in CRC cells [129]. Sulindac is a well-known non-steroidal anti-inflammatory drug (NSAID), which is an FDA-approved chemo-preventive agent that is used to inhibit adenoma development in FAP patients. We, and others, have shown that sulindac also downregulates WNT signaling by inducing the degradation of beta-catenin in CRC cells [129,130,131,132,133,134]. Thus, it is possible that the low efficacy of retinoid agents in the treatment of solid tumors is due to survivin and other IAP proteins, which become overexpressed due to *APC* mutations. Therapeutically overcoming the inhibition of apoptosis in cancer may have important relevance to the development of retinoid-based therapies against solid tumors, because the addition of arsenic trioxide to ATRA in the treatment of APL leads to synergistic effects by enhancing apoptosis [84]. Thus, the development of ATRA-based regimens for treating other cancers will likely require drug combinations that include agents such as apoptosis-inducing therapeutics.

## 9. Conclusions

The findings from our review of retinoids as chemo-preventive and molecular-targeted anti-cancer therapies reveal that RA agents continue to hold promise as effective treatments for human solid tumors. The results from the clinical trials discussed above, indicate that strategies for designing retinoid-based anti-cancer therapies will likely need to incorporate retinoids into drug combination regimens. For example, two studies [75,110] showed that RAMBAs can potentiate the effect of ATRA. Moreover, our pilot study data indicate that future therapeutic approaches will benefit, by genotyping tumors to determine the status of RA signaling genes when administering RA-based treatments to oncology patients. This is clearly exemplified by the efficacy of ATRA in APL patients whose malignant cells have a chromosome translocation, involving chimeric fusion of promyelocytic leukemia genes and the *RARA* gene. Other studies show that the RA receptor genotype is correlated with the risk for cancer development, and to the response to retinoid chemoprevention and therapy [101,135]. In this view, future research studies could also provide important information by identifying which retinoid drugs are active against tumor cells that carry specific mutations. For example, the preclinical studies discussed above suggest that RAMBA agents may have selective activity against *APC*-mutant tumor tissues. Indeed, the identification of mutations in RA receptors and other RA signaling pathway genes in human cancers, offers tremendous opportunities for target discovery, drug design, and personalized medicine for distinct molecular retinoid subtypes. Furthermore, continued discovery of the mechanisms that explain how RA signaling regulates normal tissue homeostasis, and how the dysregulation of RA signaling in cancer drives tumor growth, should provide valuable insight into how new retinoid-targeted therapies might be designed for solid tumor patients.

## Figures and Tables

**Figure 1 ijms-22-07731-f001:**
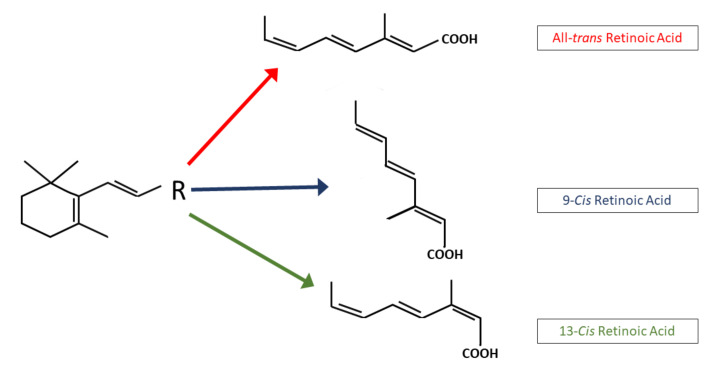
Chemical structure of physiological retinoids. The basic structure of the retinoid molecules consists of a cyclic end group, a polyene side chain and a polar end group. The conjugated system is formed by alternating C=C double bonds in the polyene side chain.

**Figure 2 ijms-22-07731-f002:**
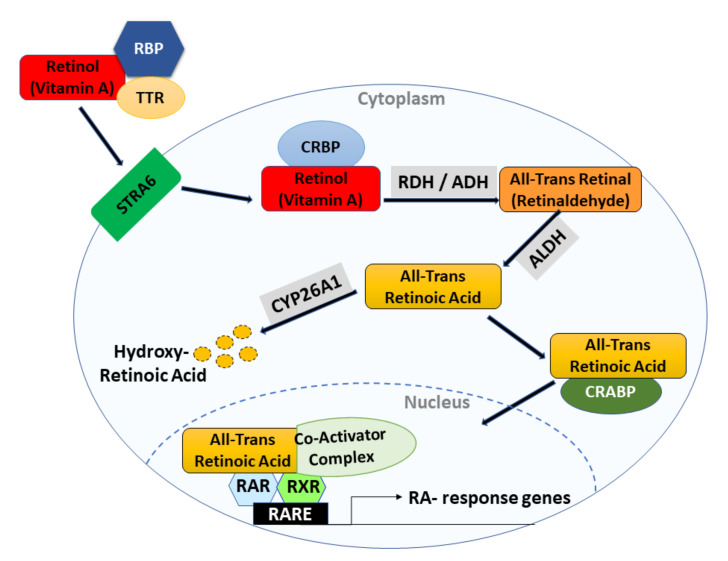
This figure illustrates the RA signaling pathway which plays a role in tissue homeostasis. Retinol, travels through the bloodstream bound to retinol-binding protein (RBP) and transthyretin (TTR). An integral plasma membrane protein, stimulated by retinoic acid 6 (STRA6), allows cellular uptake of retinol across the cell membrane into the target cell cytoplasm. Inside the cell, retinol binds to cellular retinol-binding protein (CRBP), which delivers it to the enzymes that convert it to ATRA in two steps. First, retinol is reversibly oxidized into retinal by cytosolic alcohol dehydrogenase or retinol dehydrogenase (ADH/RDH) or short-chain dehydrogenase/reductases (SDRs, not shown). Next, retinal is irreversibly converted to retinoic acid or all-trans retinoic acid by the aldehyde dehydrogenase (ALDH) enzyme family members. ATRA either gets degraded by ATRA-degrading cytochrome P450 reductases, such as CYP26A1, which converts ATRA to inactive metabolites, or it binds to cellular retinoic acid-binding proteins (CRABPs) and gets transported into the nucleus. Once inside the nucleus, ATRA functions as a ligand and binds to a heterodimer of retinoic acid receptor (RAR) and retinoid X receptor (RXR). The RAR/RXR heterodimer binds to retinoic acid response elements (RAREs) in the regulatory region of target genes, which triggers a conformational change by inducing the release of co-repressors and the recruitment of co-activator complexes.

**Figure 3 ijms-22-07731-f003:**
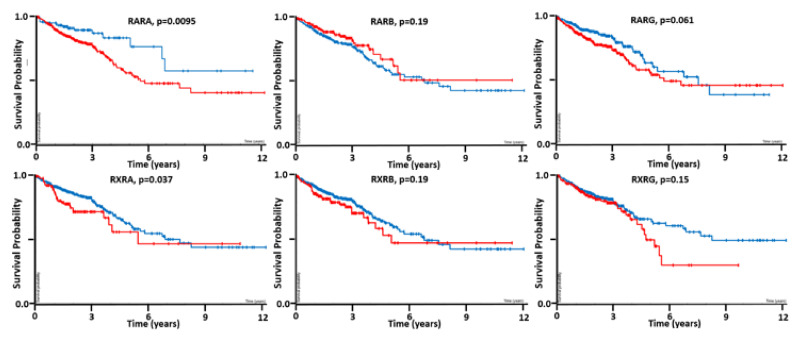
Kaplan–Meier survival analysis of RA receptor gene expression (*RARA*—**upper left** panel, *RARB*—**upper middle**, *RARG*—**upper right**, *RXRA*—**lower left**, *RXRB*—**lower middle**, *RXRG*—**lower right**) for their ability to predict CRC patient survival (red = increased; blue = decreased) [17,106]. Our bioinformatics investigation of the TCGA database [112] showed that *RARA* and *RXRA* retinoic acid receptor levels are predictive (*p* < 0.05) of CRC patient survival.

**Figure 4 ijms-22-07731-f004:**
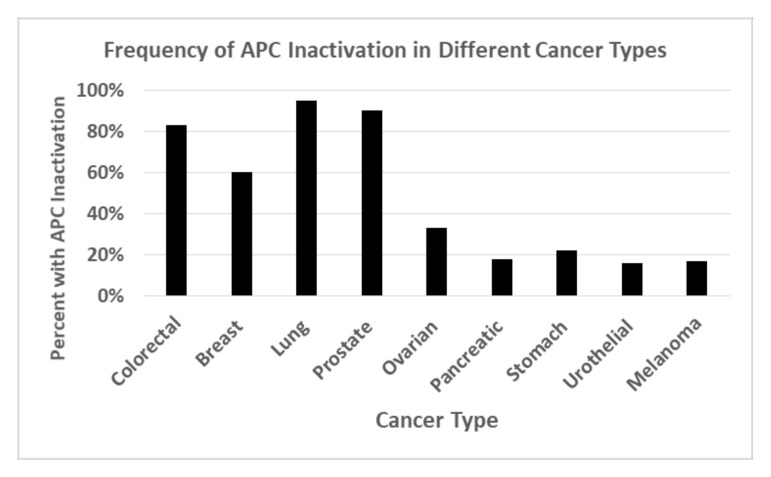
Frequency of *APC* inactivation in different human cancer types. Genomic tumor analyses show that *APC* is one of the most frequently inactivated tumor suppressor genes across all solid tumor types [116,117]. While inactivation of *APC* in FAP and sporadic CRCs is due to *APC* mutation, in other cancers, *APC* inactivation often results from hyper-methylation of APC’s promoter and its epigenetic silencing.

**Figure 5 ijms-22-07731-f005:**
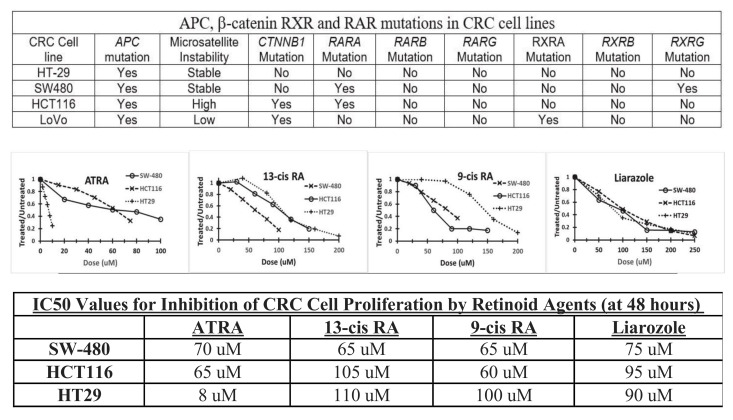
Effects of retinoid agents on CRC cells carrying RA receptor mutations. We investigated the effects of RA agents on the growth and differentiation of various human colonic cell lines (HT29, SW-480 and HCT116) [118]. The cell lines were plated, left to attach for 24 h, and serum starved for 24 h. The cells were treated for 24, 48 and 72 h with the various retinoid agents. Cell proliferation was conducted using crystal violet assay. The upper panel gives the mutation status of the CRC cell lines [17]. The middle panel shows the dose response curves for cell lines treated at 48 h. The lower panel lists the IC50 (half-maximal inhibitory concentration) values for treated cell lines at 48 h. The data reveal that the ability of various retinoid agents to induce differentiation of CSCs depends on the RA pathway genotype according to mutations in RA signaling genes in CRC cells.

**Table 1 ijms-22-07731-t001:** Clinical trial table.

Drug	Completed Clinical Trials	Current/Recruiting Clinical Trials	CT.Gov Identifier/Ref
All-Trans retinoic acid	Effect of all-trans retinoic acid with chemotherapy-based paclitaxel and cisplatin as first-line treatment of patients with advanced non-small-cell lung cancer		NCT01048645
	All-trans retinoic acid, and arsenic +/− idarubicin		NCT00413166
	A trial of all-trans retinoic acid (ATRA) in advanced adenoid cystic carcinoma		NCT03999684
		All-trans retinoic acid (ATRA) in the treatment of recurrent/metastatic adenoid cystic carcinoma of the head and neck (Aplus)	NCT04433169
	All-trans retinoic acid in combination with standard induction and consolidation therapy in older patients with newly diagnosed acute myeloid leukemia		NCT00151255
	To immunize patients with extensive stage SCLC combined with chemotherapy with or without all-trans retinoic acid		NCT00617409
		ATRA Trial—activity of ATRA in combination with anastrozole in pre-operative phase of operable HR-positive/HER2-negative early breast cancer eBC ATRA trial	NCT04113863
	Effect of all-trans retinoic acid with chemotherapy based on paclitaxel and cisplatin as first line treatment of patients with advanced non-small-cell lung cancer		NCT01048645
	All-trans retinoic acid in combination with standard induction and consolidation therapy in older patients with newly diagnosed acute myeloid leukemia		NCT00151255
	Study of dasatinib and all-trans retinoic acid for relapsed/refractory and/or elderly patients with acute myelogenous leukemia (AML) or myelodysplastic syndrome		NCT00892190
		Oral arsenic trioxide for newly diagnosed acute promyelocytic leukemia	NCT03624270
9-*Cis* retinoic acid	The pharmacokinetics of a single dose of 9-cis retinoic acid (alitretinoin, tocino) in patients with moderate-to-severe hepatic insufficiency		NCT01261923
	Alitretinoin in the treatment of chronic hand eczema		NCT00519675
	Single dose of 9-cis retinoic acid in hepatic patients		NCT01891526
	Efficacy of alitretinoin treatment in patients with pustular form of psoriasis		NCT01245140
	A phase I trial of tamoxifen and 9-cis retinoic acid in breast cancer patients		NCT00001504
	A study of ALRT1057 in patients with AIDS-related Kaposi’s sarcoma		NCT00002188
13-*Cis* retinoic acid	Identifying the genetic predictors of severe acne vulgaris and the outcome of oral isotretinoin treatment (SA)		NCT01727440
	13-cis retinoic acid with or without Vitamin E for prevention of lung cancer (13-Cis)		NCT00002586
	RA-4: 13-cis retinoic acid for treatment of men with azoospermia		NCT03323801
	Oral liquid 13-cis retinoic acid (13-CRA)		NCT03291080
	Double-blind phase III trial of effects of low-dose 13-cis retinoic acid on prevention of second primaries in stages I–II head and neck cancer		NCT03370367
	Belinostat and isotretinoin in treating patients with solid tumors that are metastatic or that cannot be removed by surgery		NCT00334789
	Isotretinoin in preventing skin cancer		NCT00025012
	Safety and effectiveness of giving isotretinoin to HIV-infected women to treat cervical tumors		NCT00001073
	Interferon alfa, isotretinoin, and paclitaxel in treating patients with recurrent small-cell lung cancer		NCT00062010
		Cabozantinib in combination with 13-cis retinoic acid in children with relapsed or refractory solid tumors.	NCT03611595
	High-dose 3F8/GM immunotherapy plus 13-cis retinoic acid for consolidation of first remission after myeloablative therapy and autologous stem cell transplantation		NCT01183416
	Isotretinoin plus interferon in treating patients with recurrent cancer		NCT00002506
	3F8/GM-CSF immunotherapy plus 13-cis retinoic acid for consolidation of first remission after non-myeloablative therapy in patients with high-risk neuroblastoma		NCT01183429
	Vorinostat and isotretinoin in treating patients with high-risk refractory or recurrent neuroblastoma		NCT01208454
	3F8/GM-CSF immunotherapy plus 13-cis retinoic acid for primary refractory neuroblastoma in bone marrow		NCT01183897
Liarozole	A randomized, double-blind, placebo-controlled phase II/III trial to evaluate the efficacy and safety of 2 doses of oral liarozole (75 mg od, and 150 mg od) given during 12 weeks in lamellar ichthyosis		NCT00282724
Talarozole	None		

## Data Availability

Data supporting the reported bioinformatics results can be found at https://www.proteinatlas.org (accessed on 15 July 2021). Information on clinical trials was obtained from the database on clinical trials listed on www.clinicaltrials.gov.

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
