# Peer review of "Retinoids as Chemo-Preventive and Molecular-Targeted Anti-Cancer Therapies"

_ijms, 2021, doi:10.3390/ijms22147731_

Round 1

Reviewer 1 Report

In this paper, the Authors review cellular signaling involving retinoids and the role of mutations in retinoic acid (RA) receptors and other RA signaling pathway genes in carcer growth and metastasis. The effects of retinoids in chemoprevention and treatment of various cancers observed in recent clinical trials (Table 1) is discussed in detail. Particular attention has been paid to the capability of retinoid agents to inhibit the growth of solid tumor cells, such as in the colorectal cancer (CRC), and in leukemia. The Authors attempt to elucidate the problem why retinoid agents generally do not show good clinical outcome against solid tumors and discuss alternative approaches that could overcome the lack of efficacy. Whereas the retinoic acid agents with anti-tumor activity are able to induce cellular differentiation, they have not been developed yet into efficient cancer  treatments. The Authors conclude that the identification of mutations in retinoid receptors and other RA signaling pathway genes may offer new opportunities for target discovery, drug design, and personalized medicine for specific retinoid subtypes. Pointing to the results of recent clinical trials, the Authors indicate that strategies for designing retinoid-based anti-cancer therapies will likely involve drug combination regimens with incorporated retinoids. Future research studies may also provide important information by identifying which retinoid drugs are active against tumor cells that carry specific mutations.

I recommend the paper for publication after minor revision addressing the issues listed below.

1. The cancer growth is usually accompanied (or stimulated) by overexpression of anti-apoptotic proteins from AIP family, such as survivin (Sur), which promote cancer cell proliferation, see for instance: Biosensors and Bioelectronics 2019, 137, 58-71. It may be possible that the low efficacy of retinoid agents in treatment of solid tumors is due to the overexpression of Sur, commonly observed in solid tumors, e.g., in colorectal cancer cells SW480 (ACS Appl. Mater. Interfaces 2018, 10, 20, 17028–17039). Any comment on this subject would benefit general Readership. These relevant references should be cited. 

Author Response

We thank the reviewers for their helpful comments. Each reviewer comment (italics) is followed by our response (â–º).

Reviewer 1 (R1) comments:

R1.1 The cancer growth is usually accompanied (or stimulated) by overexpression of anti-apoptotic proteins from AIP family, such as survivin (Sur), which promote cancer cell proliferation, see for instance: Biosensors and Bioelectronics 2019, 137, 58-71. It may be possible that the low efficacy of retinoid agents in treatment of solid tumors is due to the overexpression of Sur, commonly observed in solid tumors, e.g., in colorectal cancer cells SW480 (ACS Appl. Mater. Interfaces 2018, 10, 20, 17028–17039). Any comment on this subject would benefit general Readership. These relevant references should be cited.

â–ºThank you for this very helpful suggestion. We have added a paragraph on anti-apoptotic proteins (including the requested references) to the end of our Discussion of our paper.

Reviewer 2 Report

The authors provide a comprehensive review of the RXR/VitaminA pathway and its significance in cancer. Overall, the review highlight important findings on the mutational aspects affecting inhibitor binding that targets the pathway. However, the review covers a lot of details that would probably not necessary and keep focused on the solid tumor in general after introducing the pathways. The chemoprevention part is also needs more attention in the text.

Some comments:

Can the author include some outcome in the table? The table can show cancer type, type of VitA and outcome with trial numbers instead of the entire title of the trial for readability.

The authors could consider another table, if possible, for cell line studies and molecular mechanisms suggested with reference numbers as the last column.

Can the authors explain more on line 140? Does mutations affect co-factor binding? Please clarify.  Are the mutated RXRs expressed?

Line 167: STAT3 is well-known pro tumorigenic factors. How does ATRA affect this? Is STAT3 phosphorylation downregulated? This should be mentioned. The whole paragraph can be shortened as well unless this is a seminal study in the field.

Figures will benefit from more description. For example, Figure 3 is showing expression? How was this generated? Figure4, inactivation means mutations and the dataset used?

Can the authors provide the information about the RA receptor mutations? What type and if these are predicted to be deleterious? And type of information and mechanistic study done.

Author Response

We thank the reviewers for their helpful comments. Each reviewer comment (italics) is followed by our response (â–º).

Reviewer 2 (R2) comments:

R2.1 The chemoprevention part is needs more attention in the text.

â–ºWe have added a paragraph on chemoprevention that is included in the paper.

R2.2 Can the author include some outcome in the table? The table can show cancer type, type of Vit A and outcome with trial numbers instead of the entire title of the trial for readability.

â–ºThank you for the comment – unfortunately clinicaltrials.gov does not provide outcomes of the trials.

R2.3 The authors could consider another table, if possible, for cell line studies and molecular mechanisms suggested with reference numbers as the last column.

â–ºThese changes are well beyond the scope of our study and moreover cannot be done in 5 days within the time frame allotted by the journal editors for us to resubmit our revised manuscript. We have referenced our previous paper that gives information and molecular mechanisms for CRC cell lines.

R2.4 Can the authors explain more on line 140? Do mutations affect co-factor binding? Please clarify.  Are the mutated RXRs expressed?

â–ºWe have clarified these points in our revised manuscript.

R2.5 Line 167: STAT3 is well-known pro tumorigenic factors. How does ATRA affect this? Is STAT3 phosphorylation downregulated? This should be mentioned. The whole paragraph can be shortened as well unless this is a seminal study in the field.

â–ºWe have made the requested changes on STAT3 in our revised manuscript.

R2.6 Figures will benefit from more description. For example, Figure 3 is showing expression? How was this generated? Figure4, inactivation means mutations and the dataset used?

â–ºThank you for pointing out this deficit. We have added more detail as requested to the Figure Legends.

R2.1 Can the authors provide the information about the RA receptor mutations? What type and if these are predicted to be deleterious? And type of information and mechanistic study done.

â–ºWe have included the requested information on RA receptor mutations in our revised manuscript.